# Spatiotemporal Variations of Carbon Emissions and Their Driving Factors in the Yellow River Basin

**DOI:** 10.3390/ijerph191912884

**Published:** 2022-10-08

**Authors:** Shiqing Wang, Piling Sun, Huiying Sun, Qingguo Liu, Shuo Liu, Da Lu

**Affiliations:** 1School of Geography and Tourism, Qufu Normal University, Rizhao 276826, China; 2College of Land Science and Technology, China Agriculture University, Beijing 100193, China; 3RiZhao Key Laboratory of Territory Spatial Planning and Ecological Construction, Rizhao 276826, China

**Keywords:** carbon emissions, spatiotemporal differentiation, driving factor detection, Yellow River Basin

## Abstract

The Yellow River Basin (YRB) is a significant area of economic development and ecological protection in China. Scientifically clarifying the spatiotemporal patterns of carbon emissions and their driving factors is of great significance. Using the methods of spatial autocorrelation analysis, hot-spot analysis, and a geodetector, the analysis framework of spatiotemporal differentiation and the driving factors of carbon emissions in the YRB was constructed in this paper from three aspects: natural environment, social economy, and regional policy. Three main results were found: (1) The carbon emissions in the YRB increased gradually from 2000 to 2020, and the growth rates of carbon emissions in the different river reaches were upper reaches > middle reaches > lower reaches. (2) Carbon emissions have an obvious spatial clustering character from 2000–2020, when hot spots were concentrated in the transition area from the Inner Mongolia Plateau to the Loess Plateau. The cold spots of carbon emissions tended to be concentrated in the junction area of Qinghai, Gansu, and Shaanxi. (3) From 2000 to 2020, the driving factors of spatial differentiation of carbon emissions in the YRB and its different reaches tended to be diversified, so the impacts of socioeconomic factors increased, while the impacts of natural environmental factors decreased. The influence of the interactions of each driving factor showed double factor enhancement and nonlinear enhancement. This study will provide a scientific reference for green and low-carbon development, emphasizing the need to pay more attention to environmental protection, develop the green economy vigorously, and promote the economic cycle, so as to achieve green development and reduce carbon emissions.

## 1. Introduction

Global changes caused by greenhouse gas emissions have attracted worldwide attention in recent years. As an important participant, contributor, and leader in the response to global climate change, China has proposed carbon peak and carbon neutralization goals [1]. Since the beginning of the 21st century, due to the development of urbanization and industrialization in China, its industrial structure has adjusted continually, and carbon emissions have increased rapidly. The spatial distribution patterns of carbon emissions and their driving factors have become a hot issue in global change research, with the construction of ecological civilization rising to new heights [2]. The spatiotemporal variations and driving mechanisms of different types of carbon emissions in typical regions have become an important piece of current carbon emission research. Carbon emissions in the Yellow River Basin (YRB) have aroused widespread concern in academia. The ecological environment of the YRB is sensitive and fragile, thus becoming a hot spot for carbon emission research [3]. Especially with the implementation of China’s ecological civilization construction strategy since 2000, the YRB, as an important ecological corridor in China, has become an important region for achieving the dual carbon goal, so relevant research has been urgently needed. Therefore, clarifying the spatial patterns and driving factors of carbon emissions in the YRB is of great significance in emission reduction work, to promote ecological protection and high-quality development.

Correlation degree [4], structure decomposition [5], LMDI decomposition [6], the autoregressive distributed lag (ARDL) model [7], and the STIRPAT model [8] have been used to analyze the classification of carbon emissions [9], the spatiotemporal pattern and its driving factors [10], dynamic simulation [11], the budget and compensation [12], the decoupling effect [13], and emission reduction measures [14], but there are still some problems. These include more attention needing to be paid to the linear quantitative relationship, the spatial geographic information model being applied less often, and the currently weak research on the spatiotemporal differentiation and driving factors of carbon emissions within the region. International research on carbon emissions has focused on the spatial scales of countries, regions, provinces, cities, and counties [15,16,17,18,19,20], with the Yangtze River Delta, the Pearl River Delta, Beijing–Tianjin–Hebei, and other rapid urbanization areas as the research hot spots [21,22,23], while there are few studies on watershed areas with huge natural and economic differences. In terms of driving factors, a consensus has been reached that carbon emissions are mainly affected by urban populations, energy structures, economic development, industrial structures, and other socioeconomic factors. For instance, Gao et al. [24] found that changes in production and consumption activities had a significant impact on the spatial variation of carbon emissions in China. Santanu et al. [25] pointed out that increased fossil fuel flux led to increased carbon emissions in India. Sufyanullah et al. [26] found that there was a positive correlation between urbanization and carbon emissions in Pakistan. Van Marle et al. [27] found that land use and cover change (LUCC) had an impact on carbon emissions. Murshed et al. [28] explored that the use of nuclear energy would slow down carbon emissions, while socioeconomic growth would increase them. At present, the research on carbon emissions in the YRB mainly focuses on the following aspects: (1) Study based on different types of carbon emissions, such as energy carbon emissions, land use carbon emissions, tourism carbon emissions, and so on [29,30]; (2) Study based on spatiotemporal variation and driving factors in the view of different spatial scales, such as on the perspective of prefecture-level administrative divisions, in the view of watershed geographic differentiation, and on the analysis of county carbon emissions [10,31,32]; (3) Study on the control methods of carbon emissions, such as ecological compensation research, carbon peak levels, carbon emission reduction paths, and carbon decoupling effects [3,33,34,35]. Various methods were used in the above approaches, such as the Gini coefficient, Tapio index, spatial panel model, carbon ecological compensation model, and others [3,32,33,34]. In general, research on carbon emissions in the YRB has become increasingly perfect, providing scientific ideas for achieving carbon peak and carbon neutralization goals.

The YRB runs across all three major regions of China (East, Central, and West), where the internal natural landscape and economic development patterns are significantly different, which can be seen as a microcosm of China. As an important ecological barrier in China, the YRB plays an important role in the development of eco-environmental security. Understanding the spatiotemporal evolution and driving factors of carbon emissions in different reaches of the YRB is of great significance to promote regional high-quality development, as well as to achieve carbon peaks and carbon neutrality. Therefore, to provide a scientific basis for achieving county-level and whole-basin carbon peaks and to formulate carbon emission reduction policies, spatial autocorrelation, hotspot analysis, and a geodetector were used to analyze the spatiotemporal evolution and driving factors of carbon emissions in the YRB at a county scale. The three aims of this study are to (i) identify the temporal variation of carbon emissions in the YRB during 2000–2020; (ii) reveal the spatial differentiation of carbon emissions in different reaches; (iii) explore the factors affecting this spatiotemporal variation; and (iv) provide certain policy suggestions.

## 2. Materials and Methods

### 2.1. Study Area

The Yellow River originates from the northern Bayan Har Mountains on the Qinghai–Tibet Plateau and flows through nine provinces and regions in China. Considering the integrity of administrative units and the link between economic development and the Yellow River, the scope of the study area is defined. Taking the natural YRB as the main body, the study area located between 34°43′31″ N–46°57′46″ N and 100°57′11″ E–125°34′11″ E, with a total area of 7.95 × 10^5^ km^2^ and includes 423 counties (banners, cities, and districts) (Figure 1). The terrain is high in the west and low in the east, with elevation ranging from 20 m below sea level to 6250 m above sea level, which is dominated by plateaus. The YRB has a temperate monsoon climate, with an annual average temperature of 9 °C and annual average precipitation of 470 mm, with a climate that is sensitive [36]. At the end of 2020, the YRB had a total population of 155.74 million, with a GDP of RMB 987.44 billion. The industrial structure is dominated by secondary industries with large energy consumption. The basin is rich in energy and mineral resources, so it is not only an important energy supply and consumption area in China but also the key area of carbon emissions. As an important ecological barrier that connects the Qinghai–Tibet Plateau, the Loess Plateau, and the North China Plain, the YRB has important effect on ecological protection and high-quality development, and administrations in the region vigorously carry out energy conservation and emission reduction work, in which the contradiction between economic development and ecological and environmental protection is prominent [37].

### 2.2. Data Sources and Processing

Considering the availability, the data used in this study include carbon-emission data, nighttime lighting data, energy consumption data, meteorological data, topographic data, vegetation data, and socioeconomic data. The data sources are shown in Table 1. Partial missing data were obtained using adjacent year data substitutions or smoothing calculations.

### 2.3. Methods

#### 2.3.1. Carbon-Emission Measurement

By fitting DMSP/OLS and NPP/VIRRS nighttime lighting data using the particle swarm optimization–back propagation algorithm, we obtained data of county carbon emissions in the YRB from 2000 to 2020 based on data processing of urban greenhouse gas emissions in China, as follows [20,38]:(1)$$CO_{2i}=\sum_{i=1}^{n}\sum_{j=1}^{t}\left(E_{ij}\times LCV_{ij}\times CC_{ij}\times COF_{ij}\times \frac{44}{12}\right)$$
where *CO_2i_* is the carbon emission of county unit *i* in the YRB, n represents the number of county units, *t* is the number of energy types, *E_ij_* represents the energy use type of the county unit *i*, *LCV_ij_* represents the low calorific value of the *j* energy consumption in county unit *i*, *CC_ij_* represents the carbon content of the *j* energy consumption in county unit *i*, and *COF_ij_* represents the oxygen content of the *j* energy consumption in county unit *i*.

#### 2.3.2. Spatial Autocorrelation Analysis

Spatial autocorrelation analysis reveals the spatial distribution of a spatial element and its property value that is related to the adjacent region and the degree of correlation [39]. The spatial distribution characteristics of carbon emissions in the YRB from 2000 to 2020 were analyzed by the Global spatial autocorrelation. The spatial agglomeration characteristics of carbon emissions between local and adjacent county units in the YRB were revealed by local spatial autocorrelation [40], as follows:(2)$$I=\frac{\sum_{i=1}^{n}\sum_{j\neq i}^{n}W_{ij}\left(x_{ij}-\bar{x}\right)\left(x_{ji}-\bar{x}\right)}{s^{2}\sum_{i=1}^{n}\sum_{j\neq i}^{n}W_{ij}}$$
(3)$$I_{i}=\frac{\left(x_{i}-\bar{x}\right)\sum_{j=1}^{n}\left(x_{j}-\bar{x}\right)}{s^{2}}$$

*I* represents the global Moran’s *I* index, ranging between −1 and 1. Moran’s *I* > 0 indicates a spatial convergence trend in carbon emissions in the YRB. Moran’s *I* < 0 indicates a spatial divergence. *n* represents the number of county units; *X_i_* and *X_j_* represent the observed values of carbon emissions in spatial geographical units *i* and *j*, respectively, and are the mean values of carbon emissions in each county unit; and *W_ij_* is the space weight matrix. *I_i_* > 0 shows that there is a positive spatial correlation between adjacent county units, and the carbon emissions of adjacent county units show high–high or low–low agglomeration types. *I_i_* < 0 indicates that there is a negative spatial correlation between adjacent county units, and the carbon emissions of adjacent county units show high–low or low–high agglomeration types.

#### 2.3.3. Explanatory Variables

Spatial hot-spot detection is often used to characterize spatial clusters of a property in a local area around a spatial unit and to test the spatial correlation of a certain geographic attribute value [41]. The Getis–Ord *G_i_^*^* was used to identify the spatial cluster characteristics of carbon emissions in the YRB, as follows:(4)$$G_{i}^{*}\left(d\right)=\sum_{j=1}^{n}W_{ij}\left(d\right)X_{i}/\sum_{i=1}^{m}X_{i}$$
where *X_i_* represents the carbon emissions of county *i* in the YRB, *m* is the number of counties, and *W_ij_(d)* is the spatial weight calculated from the Euclidean distance *d* between counties *i* and *j*. If *G_i_^*^(d)* > 0, county *i* is a high-value cluster of carbon emissions (hot spot), whereas if *G_i_^*^(d)* < 0, county *i* is a low-value cluster (cold spot).

#### 2.3.4. Construction of Influencing Factor Model

(1)Analysis framework of influencing mechanism

The spatiotemporal differentiation of carbon emissions is a complex dynamic evolution process, which is the result of the interaction of natural factors and human activities. Natural factors determine the quantity and location of energy supply to a certain extent, while human activities affect the direction and degree of energy development and utilization. Therefore, the evolution of spatiotemporal pattern of carbon emissions is limited by both natural environmental factors and human activities (Figure 2).

Natural environmental factors are the geographical constraints of regional background factors such as climatic conditions and topographic conditions, which determine the basic spatial pattern of regional carbon emissions to a certain extent. Natural environmental factors, such as topography with relative stability, become the basic conditions to determine the spatial pattern of carbon emissions. Different natural environments lead to diversified coupling relationships of human activities and natural environments, which affect the spatiotemporal differentiation of carbon emissions. Topographical conditions affect the redistribution of precipitation and heat conditions, which, in turn, affect agricultural production, determining the adaptability of population distribution and industrial layout and limiting the evolution direction and occurrence probability of carbon emission. Human activities are mainly external driving force of diversified needs such as economic development and social activities, which play a key role in the spatial pattern evolution of carbon emissions. Human factors are the key factors leading to the spatiotemporal differentiation of carbon emissions. Especially, with the growth of population and the acceleration of urbanization, as we improve the economic development level, the influence of human activities on the spatiotemporal differentiation of carbon emissions increases more strongly [42]. Economic development affects the intensity of carbon emissions through economic scale, social investment, industrial structure, and residents’ income, which provides a driving force for spatiotemporal differentiation of carbon emissions [20]. Social activities, which become an important factor to promote the spatiotemporal differentiation of carbon emissions, affect the flow of urban and rural factors through population agglomeration and urbanization level. For the spatiotemporal differentiation of carbon emissions at regional scale, the formulation and implementation of regional policies play a macro-control role. Regional policies play a regulatory role in the development of regional social economy, which affect land use patterns, industrial structure layouts with land use planning, and the policy of Grain for Green. Therefore, regional policies are importance driving forces for the spatiotemporal differentiation of carbon emissions.

(2)Selection of explanatory variables

Dynamic changes of carbon emission are closely related to the natural geographical environment, socioeconomic activities, and regional policies. Taking carbon emissions as the dependent variable, this work analyzed the driving factors of the spatiotemporal differentiation of carbon emissions in the YRB from 2000 to 2020, with respect to natural environmental factors, socioeconomic factors, and policy factors (Table 2). Considering the availability of data, the annual average temperature and annual average precipitation values were used to reflect climatic conditions, so the elevation and slope were selected to characterize the topographic conditions; the population density and population urbanization rate were used to reflect the population size; and the economic density, the average social fixed-asset investment, the secondary and tertiary industry ratios, and the disposable incomes of urban residents and rural residents were selected to characterize the economic level. In addition, the change of vegetation coverage and the area of Grain for Green caused by the implementation of important strategies and ecological civilization construction projects within the basin were used to represent regional policies. The change of vegetation coverage was represented by the Normalized Differentiation Vegetation Index (NDVI), and the ecological policy was reflected by the area that changed from cultivated land to ecological land.

(3)Geographical detector

The driving factors of spatiotemporal differentiation and their interactions for carbon emissions in the YRB from 2000 to 2020 were identified by the factor detection and interaction detection functions of the geodetector. The equation is:(5)$$q=1-\frac{\sum_{h=1}^{L}N_{h}\sigma_{h}^{2}}{N\sigma^{2}}\;$$

The *q* value indicates the influence degree of driving factors on carbon emissions in the YRB, ranging from 0 to 1. The greater the *q* value is, the stronger the impact on the spatial distribution of carbon emissions. *h* = 1, 2, …, *L*, *L* is the stratification of the independent variable; *N_h_* represents the number of units in layer *h*; and *N* is the total number of units in the study area. *σ^2^* is the variance between the spatial units. Interaction detection was used to measure the interactions between driving factors of carbon emissions in the YRB, which can be divided into five types [43,44]: nonlinear weakening, single-factor nonlinear weakening, double-factor enhancement, independent, and nonlinear enhancement (Table 3).

## 3. Results

### 3.1. Temporal Variations of Carbon Emissions in the YRB

The total annual carbon emissions of the YRB and its reaches from 2000 to 2020 were calculated by Excel, and the trends were summarized (Table 4, Figure 3).

#### 3.1.1. Temporal Variations of Carbon Emissions in the Whole Basin

Figure 3 shows that the total carbon emissions in the YRB grew from 495.65 million tons to 1628.87 million tons from 2000 to 2020, and the growth rate decreased over time, showing a stabilizing trend. The total carbon emissions in the YRB decreased from 495.65 million tons to 481.73 million tons during 2000–2004, with an average annual decrease of 3.48 million tons. China began paying equal attention to economic development and environmental protection in 2000. With the implementation of a series of policies and projects, such as the western development strategy, Grain for Green, and protection of natural forests, forest land and grassland in the YRB have been restored. However, the interference of climate change and human activities resulted in the total carbon emissions decreasing slightly and then increasing slowly from 2000 to 2001. Carbon emissions in the YRB increased rapidly from 508.61 million tons in 2005 to 1544.01 million tons in 2014, with an average annual increase of 1.15 × 10^8^ tons. Chinese economy developed rapidly during this period; industrialization and urbanization accelerated in the study area along with energy consumption. Output values of secondary and tertiary industries increased from RMB 1.89 × 10^4^ billion in 2005 to RMB 7.40 × 10^4^ billion in 2014. The growth rate of carbon emissions in the YRB slowed significantly from 2015 to 2020, with total carbon emissions fluctuating between 1.57 × 10^9^ tons and 1.63 × 10^9^ tons, with an average annual increase of 1.13 × 10^6^ tons. The emission-reduction targets and tasks proposed by the Paris Agreement in 2015 influenced the carbon emissions in the YRB significantly. Since the social economy in China has transformed from rapid growth to high-quality development, and ecological civilization construction has risen to new heights, the YRB, as an important ecological corridor in China, has affected energy conservation and ecological protection significantly. The YRB is undergoing green development; the growth rate of carbon emissions has declined significantly from 2019 to 2020, and it was only 3.42% in 2020. Overall, carbon emissions in the YRB increased by 1.13 × 10^9^ tons during the study period, with an average annual increase of 5.65 × 10^7^ tons, or 11.43%. In the past 20 years, energy consumption for human activity and economic development in the YRB has been relatively large, showing an overall trend of fluctuating growth.

#### 3.1.2. Temporal Variations of Carbon Emission in Different Reaches

Figure 3 shows that the total carbon emissions in the upper reaches of the YRB fluctuated from 125.46 million tons to 490.56 million tons from 2000 to 2020, with the growth rate decreasing, which showed a convergence trend. The total carbon emissions in the upper reaches showed a downward trend from 1.25 × 10^8^ tons to 1.20 × 10^8^ tons, with an annual decrease of 5 million tons from 2000 to 2004. Carbon emissions in the upper reaches of the YRB increased rapidly from 1.24 × 10^8^ tons to 4.33 × 10^9^ tons, with an average annual increase of 3.43 × 10^8^ tons from 2005 to 2014. With the implementation of the western development strategy, the socioeconomic development level in the upper reaches increased rapidly, so carbon emissions increased. The growth rate of carbon emissions in the upper reaches slowed from 2015 to 2020, and the total carbon emissions increased from 4.31 × 10^8^ tons to 4.91 × 10^8^ tons, with an average annual increase of 1.20 × 10^7^ tons. As an important ecological function area, environmental protection of the upper reaches continued to advance with the signing of ecological compensation agreements and the construction of ecological protection leading areas. During the study period, the carbon emissions in the upper reaches increased by 3.66 × 10^8^ tons overall, with an average annual increase of 1.83 × 10^7^ tons, or 14.64%.

The total carbon emissions in the middle reaches of the Yellow River fluctuated from 2.29 × 10^8^ tons to 7.73 × 10^9^ tons from 2000 to 2020. Total carbon emissions in the middle reaches increased from 2.29 × 10^8^ tons to 2.38 × 10^8^ tons from 2000 to 2004, with an average annual increase of 9 million tons. Carbon emissions in the middle reaches increased rapidly from 2.60 × 10^8^ tons to 7.57 × 10^8^ tons during 2005 to 2014, with an average annual increase of 5.52 × 10^7^ tons. The middle reaches flow through rich energy and mineral resource-based cities that depend on energy for socioeconomic development, where associated carbon emissions are high. The overall carbon emissions in the middle reaches have shown a downward trend since 2016, and the total carbon emissions fluctuated from 7.81 × 10^8^ tons to 7.73 × 10^8^ tons, with an annual decrease of 8 million tons. With the formulation of ecological civilization policy, resource-based cities in the YRB continue to transform and weaken the dependence of economic development on energy consumption, reducing carbon emissions. Overall, carbon emissions in the middle reaches increased by 5.44 × 10^8^ tons in the study period, with average annual increase of 2.72 × 10^7^ tons, or 11.88%. 

From 2000 to 2020, the total carbon emissions in the lower reaches increased from 1.57 × 10^8^ tons to 4.02 × 10^8^ tons, which was less than the increases in the upper and middle reaches. Total carbon emissions in the lower reaches declined from 1.57 × 10^8^ tons to 1.39 × 10^8^ tons from 2000 to 2004, with an average annual decrease of 1.80 × 10^7^ tons. Carbon emissions in the lower reaches increased rapidly from 1.41 × 10^8^ tons to 3.95 × 10^8^ tons from 2005 to 2014, with an average annual increase of 2.82 × 10^7^ tons. Carbon emissions in the lower reaches fluctuated greatly from 2015 to 2020; total carbon emissions increased from 4.00 × 10^8^ tons to 4.02 × 10^8^ tons, with an average annual increase of 2 million tons. Economic development of the lower reaches started early; by relying on technological innovations to achieve economic development transformation, carbon emissions growth slowed. During the study period, the carbon emissions in lower reaches increased by 2.45 × 10^8^ tons, with average annual increases of 2.72 × 10^7^ tons, or 7.80%.

### 3.2. Spatial Pattern Evolution of Carbon Emissions in the YRB

#### 3.2.1. Overall Characteristics of Carbon Emissions

The broad trends in carbon-emissions distribution in the YRB in 2000, 2010, and 2020 were analyzed by trend analysis tool of ArcGIS. Taking the value of the carbon emissions as the height attribute (Z value), a spatial visualization map was obtained (Figure 4). In Figure 4, green lines represent the east–west distribution of carbon emissions and blue lines represent the north–south distribution of carbon emissions. Overall, there was significant spatial differentiation in carbon emissions, which were high in the east and low in the west as well as high in the middle and low in the north and south, forming an inverted U-shaped differentiation from north to south. From 2010 to 2020, the carbon emissions of the YRB showed an overall pattern of high in the north and east and low in the west and south. The east–west and north–south directions had roughly linear distributions, in which the carbon emissions of Inner Mongolia in the north and Shandong and Henan in the east were the largest, followed by Shanxi, Shaanxi, Ningxia, and other provinces in the middle; Qinghai and Sichuan were the lowest.

#### 3.2.2. Spatial Agglomeration Characteristics of Carbon Emissions

The global Moran’s I values for carbon emissions in the YRB from 2000 to 2020 were calculated using ArcGIS (Table 5). The results were 0.249, 0.239, and 0.210 in 2000, 2010, and 2020, respectively, and all passed the significance test (Z > 1.96, *p* < 0.01), indicating that there were significant spatial correlations of carbon emissions, while the spatial clusters decreased from 2000 to 2020 and showed a convergence trend.

The spatial distribution of carbon emissions in the YRB from 2000 to 2020 was analyzed by ArcGIS (Figure 5). From 2000 to 2020, the total carbon emissions in the YRB and its different reaches have increased significantly. Carbon emissions rely on capital cities along the main stream and tributaries of the Yellow River, including Yinchuan, Taiyuan, Xian, Zhengzhou, Jinan, and major industrial cities including Baotou, Yulin, Ordos, Datong, and Zibo, showing a trend of agglomeration. In 2020, a high-value center of carbon emissions formed at the ‘ji’ bay of the Yellow River, mainly including Lingwu, Yijinhuoluo, Zhungeer, Dongsheng, Jiuyuan, and other counties in the border areas of Ningxia, Inner Mongolia, and Shaanxi.

The local spatial autocorrelation module of ArcGIS was used to calculate the local indexes of carbon emissions in the YRB from 2000 to 2020, and spatial cluster maps were drawn (Figure 6). From 2000 to 2020, the hot spots in the middle and upper reaches of the YRB expanded to the southern Loess Plateau, while those in the lower reaches concentrated in the coastal areas; cold spots were confined to the southwest.

In 2000, hot spots were concentrated in Inner Mongolia and Ningxia of the upper reaches and Shandong Henan of the lower reaches. The hot spots in the upper reaches involved 18 counties (cities and districts) and in the lower reaches involved 48 counties (cities and districts) such as Fengqiu, Yanjin, Dongping, and Zouping (Figure 6). Moreover, there were three small-scale hot spots in the middle reaches in Shanxi Province. Cold spots of carbon emissions were concentrated in Qinghai, Gansu, and Shaanxi provinces, including 136 counties (cities and districts) of Yongjing, Guide, Taibai, and Heshui, which are at the transition from the Qinghai–Tibet Plateau to Loess Plateau. Subhot spots surrounded the hot spots, forming at the junctions of the middle–upper reaches and the middle–lower reaches. Subcold spots surrounded the cold spots, including 28 counties such as Tianjun, Xiji, Songpan, Luochuan, and others.

Carbon emissions in the YRB formed two hot spots in 2010, in the middle–upper reaches and the lower reaches; the hot spots in the middle–upper reaches expanded significantly compared with 2000 and were distributed in Ningxia, Gansu, Inner Mongolia, Shaanxi, and other provinces, including 46 counties of Dongsheng, Wuyuan, Pingluo, Fugu, and others, while hot spots in the lower reaches were concentrated along the border of Shandong and Henan provinces, including 43 counties (cities and districts). The cold spots expanded to the northwest compared with 2000, including 145 counties (cities and districts) such as Gangu, Pengyang, Qianyang, and Aba, while subhot spots and subcold spots shrank (Figure 6b).

Two hot spot clusters occurred in 2020, at the junction of the middle–upper reaches and in the lower reaches in Shandong Province. The hot spots at the junction of the middle–upper reaches had expanded, compared with 2010, and were distributed in Ningxia, Inner Mongolia, Shanxi, and Shaanxi provinces to include 60 counties (cities, districts, and banners), such as Hangjin Banner, Yuyang District, Tongxin County, and Wuzhai County. Hot spots in the lower reaches were all located in Shandong Province, including 38 counties (cities, districts) such as Boxing County, Zichuan District, Liangshan County, and others. Cold spots shrank in the south compared with 2010, including 139 counties (cities, districts) such as Wanrong, Yongshou, Lingtai, Longde, and others. The subhot spots shrank, while the subcold spots expanded.

Overall, the carbon emissions in the YRB from 2000 to 2020 have obvious polarization and spatial cluster characteristics. The hot spots were concentrated in the transition area from the Inner Mongolia Plateau to the Loess Plateau in the middle–upper reaches and in the lower reaches, while the cold spots were concentrated in the Qinghai–Tibet Plateau and in the Loess Plateau hilly area, at the junction of Qinghai, Gansu, and Shaanxi.

### 3.3. Driving Factors of Spatiotemporal Variations in Carbon Emissions in the YRB

#### 3.3.1. Driving Factor Detection

Driving factors of spatiotemporal variation of carbon emissions in the YRB were identified using the geodetector (Table 6). The spatiotemporal differentiation characteristics of carbon emissions in the YRB and its different reaches from 2000 to 2020 were affected by natural environmental, socioeconomic, and policy factors.

(1)Natural environmental factors

The natural environmental factors affecting carbon emissions in the whole YRB in 2000 were elevation (X3) > slope (X4) > annual average temperature (X1) > annual average precipitation (X2). The natural environmental factors in the upper reaches were elevation > slope > annual average temperature > annual average precipitation; the natural environmental factor that affected carbon emissions more in the middle reaches was slope, while the lower reaches it was annual average temperature > annual average precipitation. These results indicate that climatic and topographic conditions are the macroscale background of agricultural production and economic development in the YRB, which determine the population and industrial layout, in turn leading to the spatiotemporal differentiation of carbon emissions. The impacts of annual average precipitation and slope on carbon emissions in the YRB increased from 2000 to 2020, while the impacts of annual average temperature and elevation decreased. The overall impact of natural environmental factors on carbon emissions in the upper reaches decreased. The impacts of annual average temperature and annual average precipitation on carbon emissions in the middle reaches increased, while the impact of slope decreased. The impacts of annual average temperature and annual average precipitation on carbon emissions in the lower reaches decreased. The natural conditions of the upper, middle, and lower reaches of the YRB are different, resulting in significant differences in carbon emissions and driving factors. The upper reaches are mainly distributed on the boundary between the Qinghai–Tibet Plateau and the Loess Plateau, where the ecological environment is fragile and sensitive, and the socioeconomic development is greatly affected by natural environmental factors. The middle–lower reaches have few differences in the natural environments with high level of economic development, so the influences of natural conditions on carbon emissions are not obvious.

(2)Socioeconomic factors

Table 4 shows that the socioeconomic factors affecting carbon emissions in the YRB overall in 2000 were population density (X5) > the tertiary industry ratio (X10), while the impact of others were not significant; the socioeconomic factors with impacts on the upper reaches were population urbanization rate (X6) > population density; the factor with the most impact on carbon emissions in the middle reaches was population density, and the socioeconomic factors that had strong impacts on lower reaches carbon emissions were population urbanization rate > population density. The impacts of population urbanization rate, economic density, average social fixed-asset investment, secondary industry ratio, tertiary industry ratio, and the average income of urban and rural residents on carbon emissions in the YRB increased from 2000 to 2020, while the impact of population density decreased. The impacts of population density, economic density, average social fixed-asset investment (X8), secondary industry ratio (X9), tertiary industry ratio, and the average income of urban (X11) and rural residents (X12) on carbon emissions in the upper reaches increased from 2000 to 2020, while the impact of population urbanization rate decreased. In the middle reaches, the impacts of population urbanization rate, economic density, secondary industry ratio, tertiary industry ratio, and the average income of urban and rural residents on carbon emissions increased, while the impact of population density decreased. The impacts of population density, population urbanization rate, economic density, average social fixed-asset investment, tertiary industry ratio, and the average income of urban residents on carbon emissions increased in the lower reaches. In the early stages of socioeconomic development, the differences in economic development levels between regions were small, as were the influences of socioeconomic factors on carbon emissions. With the acceleration of urbanization and industrialization and the rapid development of the social economy, the imbalance of development between regions increased, and the influences of socioeconomic factors on the spatial distribution of carbon emissions were enhanced.

(3)Policy factors

The policy factor most affecting carbon emissions in the YRB overall in 2000 was ecological policy, while the policy factor that had the strongest impact on carbon emissions in the upper reaches and middle reaches was vegetation coverage (X13), and the impacts of policy factors on carbon emissions in the lower reaches were not obvious. From 2000 to 2020, the impact of vegetation coverage on carbon emissions in the YRB increased, and the impact of ecological policies (X14) decreased, while the impact of vegetation coverage on carbon emissions in the upper reaches and middle reaches decreased, and the impact of ecological policies on carbon emissions in the lower reaches increased. The functional situations of the upper, middle, and lower reaches of the YRB are different, and the policy implementations and their impacts on carbon emissions are also different. The upper reaches focused on strengthening water conservation capacity and desertification control; economic development was slow in the early stage, and the binding force of policies decreased with the rapid economic development in the later stage. The middle reaches flow through resource-based cities; as they transformed, the influence of policy factors on carbon emissions increased. The impacts of policy factors decreased in the lower reaches after the completion of urban transformation and the construction of ecological projects such as Grain for Green. With the greening of Taihang in the lower reaches, the coverage rate of forest land has increased significantly, affecting the spatial distribution of carbon emissions.

#### 3.3.2. Results of Interaction Detection

The effects of interactions among driving factors on the spatiotemporal differentiation of carbon emissions in the YRB from 2000 to 2020 were detected by the interaction module of the geodetector, and the top 10 factors were selected to analyze the interaction effect of factors (Table 7).

The effects of interactions among the different driving factors were enhanced by double-factor enhancement and nonlinear enhancement. The interactions of the factors affecting carbon emissions in the YRB overall in 2000 were dominated by combinations of natural environmental factors and socioeconomic factors, indicating that the interactions between natural environmental factors and socioeconomic factors in this period enhanced the explanatory power of each driving factor on the spatial differentiation of carbon emissions. Concurrently, the interactions between natural environmental factors, socioeconomic factors, and policy factors in the upper reaches of the YRB enhanced the explanatory power of each driving factor of carbon emissions. The combinations of factors that impacted carbon emissions in the middle reaches and lower reaches were the combinations of population density with natural environmental factors and the combinations of socioeconomic factors, respectively. Meanwhile, the interactions between socioeconomic factors and policy factors had strong impacts on the carbon emissions in the upper reaches, and the combinations of socioeconomic factors had strong effects on carbon emissions in the middle reaches. The interactions between socioeconomic factors and vegetation coverage and the interactions among socioeconomic factors had obvious impacts on the spatial differentiation of carbon emissions in the lower reaches.

The interactions of natural environmental factors, policy factors, and economic density of the YRB in 2020 enhanced the explanatory power of each driving factor on the spatial differentiation of carbon emissions. Meanwhile, the combinations of factors that had significant impacts on the upper reaches and middle reaches were the combinations of natural environmental factors with socioeconomic factors and the combinations of socioeconomic factors. The combinations of population density, economic density, and other socioeconomic factors had significant impacts on carbon emissions in the lower reaches.

In summary, the interactions of natural environmental factors, socioeconomic factors, and policy factors in the YRB enhanced the spatial differentiations of carbon emissions from 2000 to 2020. Among them, annual average temperature ∩ population density, elevation ∩ area of returning cultivated land, slope ∩ population density, and slope ∩ population density had the most significant effects on the spatial differentiation of carbon emissions in the whole basin, upper reaches, middle reaches, and lower reaches in 2000, respectively. The most significant effects were annual average temperature ∩ economic density, population urbanization rate ∩ second industry ratio, population density ∩ population urbanization rate, and social fixed-asset investment ∩ vegetation coverage in 2010. Annual average precipitation ∩ economic density, slope ∩ economic density, annual average precipitation ∩ economic density, and population density ∩ social fixed-asset investment had the most significant effects on the spatial differentiation of carbon emissions in the whole basin, upper reaches, middle reaches, and lower reaches in 2020, respectively.

## 4. Discussion

### 4.1. Spatiotemporal Variation of Carbon Emissions in the YRB

Affected by natural conditions, socioeconomic development, and regional policies, the carbon emissions in the upper, middle, and lower reaches of the YRB have changed dramatically. Similar conclusions appeared in Wang et al.’s [45] study, which reported that high carbon emissions in China were mainly concentrated in the cities of eastern and central China and the central and northern regions of Inner Mongolia. Liu, Shao, and Ji [46] identified a spatial convergence model of carbon emissions in the county units of China, which had significant positive spatial correlations, also proving our point of view to some extent. However, the methods used in these studies and the measures of their impact on carbon emissions are also different from ours. Yu, Zhang, and Shen [47] reported that the intensity of carbon emissions in counties of China declined, and the Moran’s *I* index increased from 2009 to 2017. Li, Wang, and Yang [48] found that the intensity of carbon emissions of the counties in Hunan Province showed a downward trend. These results were different from ours, which is because the scale of the study area the level of regional development, and the formulation of carbon emission policies are different. Moreover, Wu et al. [49] pointed out that the growth rate of carbon emissions in the east of China was significantly higher than that in center and west, which was different from the results found in this study: the growth rate of carbon emissions in the YRB was higher in the upper reaches than in the middle or lower reaches. The differences are caused by the different measurement indicators of carbon emissions that were selected from the different research perspectives. Moreover, the focus of the above research on carbon emissions is also different from ours, as it mainly used a random forest model, the carbon conversion coefficient method, and so on to analyze the economic factors that are affecting China’s carbon emissions and the factors affecting agricultural carbon emissions, while our research systematically analyzed the natural environmental factors, socioeconomic factors, and regional policy factors that affect the spatiotemporal distribution pattern of carbon emissions in the YRB.

### 4.2. Driving Factors of Carbon Emissions in the YRB

The driving factors of carbon emissions in the YRB and its reaches were analyzed with the geographical detector. The results showed that the impact of natural environmental factors on carbon emissions in the YRB and its reaches weakened from 2010 to 2020, while the impact of socioeconomic factors such as urbanization rate, secondary industry ratio, tertiary industry ratio, GDP, and social fixed-asset investment increased significantly, which were consistent with the results of Zhou, Li, and Zhang [50] and Liu and Liu [51], indicating that our research has certain credibility. However, the above research mainly focused on the analysis of agricultural carbon emissions, while our research conducted an overall analysis of all types of carbon emissions and explored its driving factors. Compared with natural environmental factors and socioeconomic factors, policy factors had less of an impact on carbon emissions in the YRB during the study period, which was different from the conclusion of Zhang and Zhang [52], who found that ecological protection policies have a significant effect on carbon emissions. Wang and Zeng [53] reported that in different equilibrium situations of the tripartite stakeholder game, local governments and third-party verification institutions played a leading role in the strategic choices of carbon emission in different stable equilibrium circumstances. Therefore, the impact of policies is different with different equilibrium situations of different regions, and it is difficult to effectively achieve carbon emission reduction only by relying on government constraints. The areas converted from cultivated land to ecological land were used to characterize the impact of the Grain for Green policy in this work and showed that the policy has little impact on carbon emissions in the YRB, which was different from the result of Deng et al. [54], who found that the conversion of agricultural land to forest land and grassland effectively reduced carbon emissions in the Beijing–Tianjin–Hebei region. This is due to the different positioning of regional development and different policy implementation. Meanwhile, the above research mainly used linear research methods to explore the impact of a single factor, such as heavy industry and land use on carbon emissions, while our research used a spatial geographic information model to systematically analyze the natural environmental factors, socioeconomic factors, and regional policy factors that affect carbon emissions. In addition, there are differences between the central government and local governments in policy formulation and implementation, so the impact of policies on carbon emissions in different regions is also different.

### 4.3. Limitations and Prospects

Limited by the lag of statistical data publication in China, most socioeconomic data and energy consumption data in the study area are only updated to 2020, while only a few counties are updated to 2021, and most relevant data are not available in 2022. Thus, the study can only be updated to 2020. However, by studying the spatiotemporal distribution characteristics and their driving factors of carbon emissions in the study area from 2000 to 2020, regular conclusions can be obtained, which are of certain significance to guide carbon emission reduction.

Although only the YRB is selected as the research area in this study, the region is very representative and has important research significance in China. The differences of carbon emission distribution and their driving factors in the upper, middle, and lower reaches of the YRB can reflect the differences between the eastern, middle, and western regions of China. By combing the relevant research on carbon emissions in the YRB, we found that the existing research has paid little attention to the spatiotemporal distribution differences of carbon emissions between the different reaches of the YRB and between different counties in the same reach, as well as that the interaction analysis between natural environment, social economy, and regional policy factors is not enough. Based on the analysis of the spatial evolution of carbon emissions in the YRB during the past 20 years, this study focused on the effect of the interaction between different factors on the spatiotemporal differentiation of carbon emissions in the study area, to provide a scientific reference for carbon emission reduction and carbon peaking.

The results showed that policy factors have less of an impact on carbon emissions, so different carbon emission reduction policies should be put forward for different regions of the YRB, and the intensity of policy implementation should be strengthened. For example, the high value areas of carbon emissions in the YRB are concentrated in industrial cities and provincial capitals; for industrial cities such as Baotou, Yulin, Datong, and Zibo, the key way is to adjust the industrial structure and optimize the energy structure, while for capital cities such as Taiyuan, Xi’an, Zhengzhou, and Jinan, we should adhere to the development of energy-saving industries and create green low-carbon industrial parks. We found that the distribution of carbon emissions in the YRB has obvious spatial heterogeneity. Therefore, more attention should be paid to coordinated development in the basin to narrow the gaps between regions in economic development, population density, industrial structure, and carbon emissions and to work together to achieve ecological protection and high-quality development. During the study period, carbon emissions continued to increase in the YRB. Accordingly, the YRB should not only ensure the stable development of regional economy but also implement relevant supporting policies such as a stepped-carbon tax, paid allocation, industrial transfer policy, and reward policy for different regions. Meanwhile, the supervision and evaluation of carbon emission reduction policies from the national and regional levels and from the three dimensions of policy objectives, policy instruments, and policy implementation are needed to promote the realization of green and low-carbon development in the future. In addition, on account of carbon emissions having great inertia for economic development, which makes it difficult to achieve rapid reductions in the short term, the region should pay attention to environmental protection, develop the green economy vigorously, and promote the economic cycle, so as to achieve green development and reduce carbon emissions. It is also important to build an incentive mechanism of the emission reduction policy, which is led by the fiscal system and coordinated by the financial system, to give full play to the important role of finance in supporting carbon emission reduction and promoting the optimization and upgrade of energy structures and industrial structures in future development, to achieve the win–win situation of economic development and carbon-emission reduction.

## 5. Conclusions

Based on spatial autocorrelation, hot-spot analysis, and a geographical detector, this work systematically analyzed the spatiotemporal variations of carbon emissions and their driving factors in the YRB from 2000 to 2020. We found that the carbon emissions of the study area increased from 495.65 million tons to 1628.87 million tons during the study period, and the growth rates of carbon emissions in the different reaches showed the trend of upper reaches > middle reaches > lower reaches, with average annual growth rates of 16.81%, 10.28%, and 7.77%, respectively. From 2000 to 2020, the carbon emissions in the YRB have obvious spatial-cluster characteristics at the county level. The hot spots of carbon emissions are in the Inner Mongolia Plateau and Loess Plateau, at the border of the middle–upper reaches and in the lower reaches, whereas the cold spots are in the mountainous area of the Qinghai–Tibet Plateau and the Loess Plateau at the junction of Qinghai, Gansu, and Shaanxi. Spatiotemporal variations of carbon emissions were influenced by natural environmental, socioeconomic, and policy factors. The impacts of annual average precipitation, slope, population urbanization rate, economic density, social fixed-asset investment, secondary industry ratio, tertiary industry ratio, and the average income of urban and rural residents increased, while the impacts of annual average temperature, elevation, population density, vegetation coverage, and the Grain for Green policy decreased. The driving factors in the YRB and its different reaches tended to be diversified, and the influences of the interactions of each driving factor on the spatiotemporal differentiations of carbon emissions showed double-factor and nonlinear enhancement effects.

## Figures and Tables

**Figure 1 ijerph-19-12884-f001:**
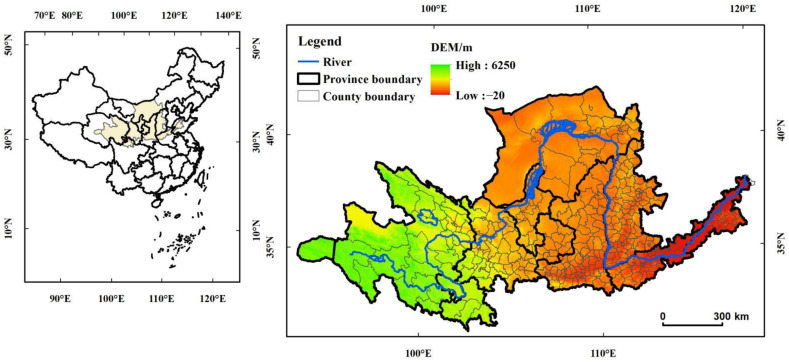
Location of study area.

**Figure 2 ijerph-19-12884-f002:**
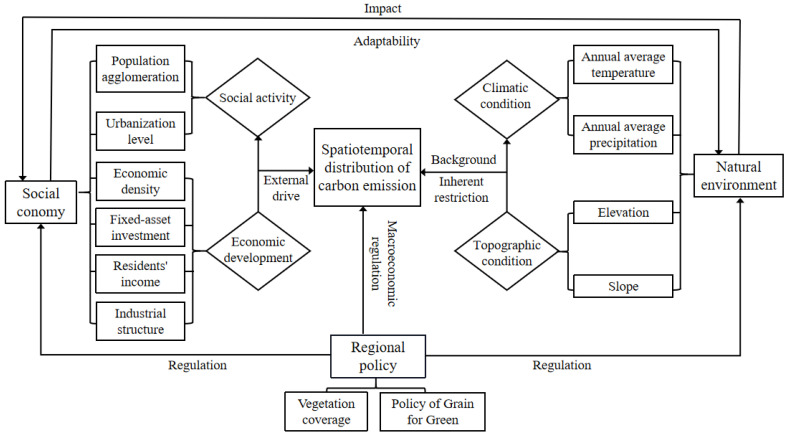
The analysis framework of influencing mechanism.

**Figure 3 ijerph-19-12884-f003:**
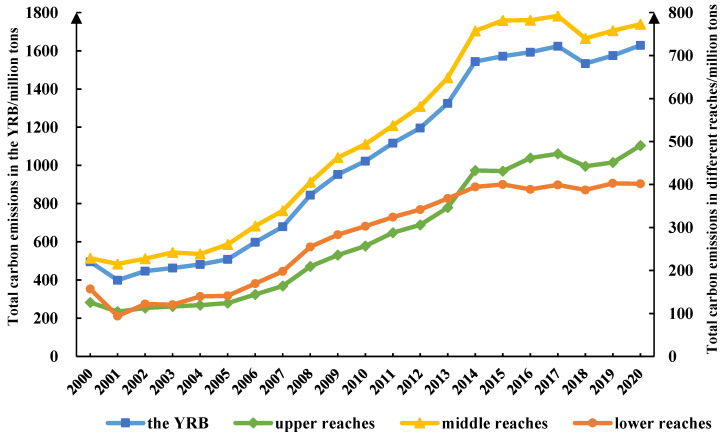
Trends of carbon emissions in the YRB during 2000–2020.

**Figure 4 ijerph-19-12884-f004:**
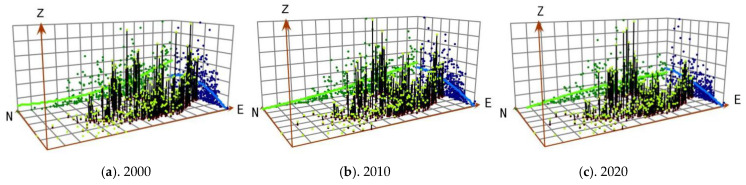
Distribution trends of carbon emissions in the YRB in (**a**) 2000, (**b**) 2010, and (**c**) 2020.

**Figure 5 ijerph-19-12884-f005:**
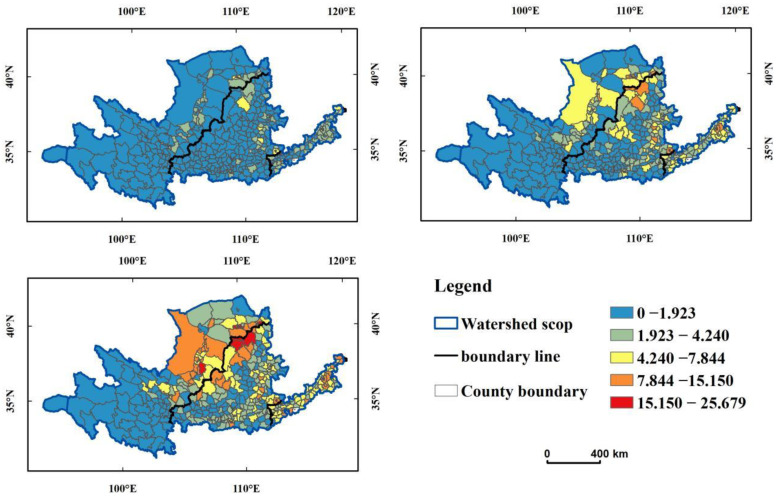
Distribution trend of carbon emissions in different streams of the YRB.

**Figure 6 ijerph-19-12884-f006:**
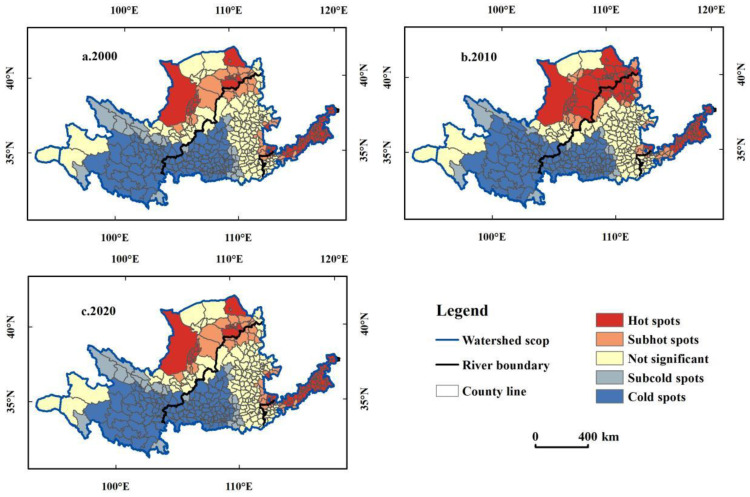
Distribution of carbon emissions hot and cold spots in the YRB: (**a**) 2000, (**b**) 2010, and (**c**) 2020.

**Table 1 ijerph-19-12884-t001:** Data type and sources.

Data Type	Data Sources
Carbon emissions data	China Carbon Accounting Database (https://www.ceads.net/), (accessed on 1 May 2022)
Defense Meteorological Program/Operational Line-Scan System (DMSP/OLS; 2000–2013)	Resource and Environmental Science and Data Center (https://www.resdc.cn/), (accessed on 1 May 2022)
Suomi National Polar-Orbiting Partnership/Visible Infrared Imaging Radiometer Suite (Suomi NPP/VIRRS; 2013–2020) nighttime lighting data	National Geophysical Data Center (http://www.geodata.cn/), (accessed on 1 May 2022)
Meteorological data	China Meteorological Science Data Sharing Service Network (http://cdc.cma.gov.cn/), (accessed on 1 May 2022)
Socioeconomic statistics	Statistical Yearbooks, Statistical Communiques, and China County Statistical Yearbooks of the provinces and regions in the YRB

**Table 2 ijerph-19-12884-t002:** Explanatory variables relevant to spatiotemporal variation of carbon emissions.

Driving Factor	Explanatory Variable	Impact Factor	Interpretation
Natural environmental factors	Climate condition	X1 Annual average temperature (°C)	Value of each unit obtained by the method of Kring with ArcGIS software
		X2 Annual average precipitation (mm)	Value of each unit obtained by the method of Kring with ArcGIS software
	Topographic condition	X3 Elevation (m)	Digital elevation map (DEM) data of all counties (cities, districts and flags) in the Yellow River Basin (YRB) obtained by analysis tool with ArcGIS software
		X4 Slope (°)	Average slope of each regional unit extracted based on DEM data
Socioeconomic factors	Population size	X5 Population density (people/km^2^)	Total population divided by the total regional area
		X6 Population urbanization rate (%)	Proportion of nonagricultural population in all regions
	Economic level	X7 Economic density (×10^9^ RMB/km^2^)	Gross domestic product divided by the total regional area
		X8 Average fixed-asset investment (×10^8^ RMB/km^2^)	Fixed-asset investment divided by the total regional area
		X9 Second industry ratio (%)	Proportion of secondary industries
		X10 Tertiary industry ratio (%)	Proportion of tertiary industries
		X11 Disposable income of urban residents (RMB)	Disposable income of urban residents in each regional unit
		X12 Disposable income of rural residents (RMB)	Disposable income of rural residents in each regional unit
Regional policy factors	Vegetation coverage	X13 Normalized Differentiation Vegetation Index (NDVI)	Normalized vegetation index obtained by spatial interpolation method with ArcGIS
	Policy of Grain for Green	X14 Area of returning cultivated land (km^2^)	Conversion area extracted from cultivated land to ecological land (forest land, grassland, water area) with ArcGIS

**Table 3 ijerph-19-12884-t003:** The results of interaction types.

Interaction Types	Condition
nonlinear weakening	q(X1∩X2) < Min(q(X1),q(X2))
single-factor nonlinear weakening	Min(q(X1),q(X2)) < q(X1∩X2) < Max(q(X1),q(X2))
double-factor enhancement	q(X1∩X2) > Max(q(X1),q(X2))
independent	q(X1∩X2) = q(X1)+q(X2)
nonlinear enhancement	q(X1∩X2) > q(X1)+q(X2)

**Table 4 ijerph-19-12884-t004:** Total carbon emissions and changes in the YRB and its reaches during 2000–2020.

Region	Carbon Emissions in 2000/Million Tons	Carbon Emissions in 2010/Million Tons	Carbon Emissions in 2020/Million Tons	2000–2020	
				Variation/Million Tons	Change Rate/%
Whole basin	495.65	1023.03	1628.87	1133.22	228.64
Upper reaches	147.89	324.93	645.08	497.19	336.18
Middle reaches	190.45	395.24	582.30	391.58	205.60
Lower reaches	157.30	302.87	401.76	244.46	155.41

**Table 5 ijerph-19-12884-t005:** Global Moran’s I of carbon emissions in the YRB in 2000, 2010, and 2020.

Year	Global Moran’s *I*	*E(G_i_^*^)*	*Z(G_i_^*^)*	*p*
2000	0.249	–0.004	8.952	0.000
2010	0.239	–0.004	9.090	0.000
2020	0.210	–0.004	8.818	0.000

**Table 6 ijerph-19-12884-t006:** Detection factor q for carbon emissions in the YRB from 2000 to 2020.

Impact Factor	2000				2010				2020			
	Whole Basin	Upper Reaches	Middle Reaches	Lower Reaches	Whole Basin	Upper Reaches	Middle Reaches	Lower Reaches	Whole Basin	Upper Reaches	Middle Reaches	Lower Reaches
X1	0.140 ***	0.189 ***	0.017	0.222 ***	0.111 ***	0.162 ***	0.022	0.179 **	0.123 ***	0.132 ***	0.056 *	0.147 **
X2	0.067 ***	0.192 ***	0.031	0.212 ***	0.109 ***	0.432 ***	0.091	0.077 **	0.087 ***	0.180 ***	0.103 ***	0.036
X3	0.188 ***	0.344 ***	0.039	0.005	0.173 ***	0.381 ***	0.014	0.001	0.130 ***	0.111 ***	0.017	0.002
X4	0.171 ***	0.233 ***	0.092 *	0.063	0.181 ***	0.326 ***	0.074	0.063	0.180 ***	0.120 ***	0.060	0.044
X5	0.212 ***	0.133 ***	0.246 ***	0.222 *	0.197 ***	0.098	0.221 ***	0.336 ***	0.133 ***	0.159 **	0.185 ***	0.268 ***
X6	0.009	0.212 ***	0.017	0.131	0.068 ***	0.244 ***	0.133 ***	0.041	0.121 ***	0.087 ***	0.137 ***	0.177 **
X7	0.099	0.007	0.001	0.305	0.430 ***	0.539 ***	0.296 ***	0.396 ***	0.297 ***	0.591 ***	0.292 ***	0.368 **
X8	0.010	0.001	0.052	0.026	0.387 ***	0.550 ***	0.219 ***	0.497 ***	0.113 ***	0.334 ***	0.084	0.252 *
X9	0.055	0.004	0.007	0.161	0.365 ***	0.532 ***	0.271 ***	0.118	0.193 ***	0.428 ***	0.189 **	0.228
X10	0.063 *	0.001	0.013	0.164 *	0.348 ***	0.424 ***	0.171 *	0.266 *	0.214 ***	0.452 ***	0.100 *	0.290 *
X11	0.001	0.049	0.001	0.019	0.025 **	0.068 **	0.031	0.005	0.160 ***	0.448 ***	0.088 **	0.092
X12	0.001	0.001	0.005	0.019	0.181 ***	0.205 ***	0.235 ***	0.139 **	0.163 ***	0.370 ***	0.139 ***	0.180 **
X13	0.013	0.211 ***	0.087 *	0.147	0.060 **	0.255 ***	0.104*	0.047	0.104 ***	0.053 ***	0.077	0.125
X14	0.050 ***	0.057	0.004	0.035	0.016	0.090 **	0.011	0.222 **	0.015	0.049	0.001	0.110 **

Note: ***, **, and * indicate significant correlations at the 0.01, 0.05, and 0.10 levels, respectively.

**Table 7 ijerph-19-12884-t007:** Results of interactions among the influencing factors from 2000 to 2020.

Year	* Whole Basin		Upper Reaches		Middle Reaches		Lower Reaches	
	Interactive Factors	Interactive Value	Interactive Factors	Interactive Value	Interactive Factors	Interactive Value	Interactive Factors	Interactive Value
2000a	X1∩X5	0.359 *	X3∩X14	0.484 *	X4∩X5	0.456 *	X4∩X5	0.622 *
	X4∩X5	0.351 **	X4∩X6	0.471 **	X13∩X5	0.422 *	X1∩X5	0.562 *
	X4∩X3	0.347 **	X2∩X3	0.463 **	X1∩X5	0.406 *	X13∩X5	0.557 *
	X1∩X3	0.340 *	X13∩X3	0.441 **	X2∩X5	0.369 *	X13∩X4	0.538 *
	X13∩X5	0.336 *	X1∩X3	0.440 **	X6∩X5	0.362 *	X7∩X5	0.536 **
	X2∩X5	0.324 *	X13∩X6	0.423 **	X3∩X5	0.329 *	X6∩X5	0.531 *
	X3∩X5	0.312 **	X3∩X5	0.403 **	X8∩X5	0.265 **	X9∩X13	0.510 *
	X2∩X3	0.300 *	X1∩X6	0.398 **	X12∩X5	0.265 *	X9∩X5	0.502 *
	X1∩X2	0.295 *	X6∩X14	0.392 *	X2∩X6	0.253 *	X13∩X7	0.496 *
	X1∩X4	0.291 **	X13∩X4	0.386 **	X11∩X5	0.252 **	X10∩X5	0.490 *
2010a	X1∩X7	0.550 *	X9∩X6	0.725 **	X6∩X5	0.555 *	X13∩X8	0.695 *
	X2∩X7	0.548 *	X2∩X7	0.724 **	X6∩X7	0.532 *	X13∩X7	0.653 *
	X6∩X7	0.541 **	X13∩X7	0.721 **	X6∩X8	0.507 *	X8∩X5	0.652 **
	X4∩X7	0.536 **	X6∩X7	0.718 **	X9∩X5	0.503 **	X10∩X7	0.639 **
	X10∩X7	0.532 **	X14∩X7	0.707 *	X12∩X7	0.485 **	X8∩X7	0.604 **
	X13∩X7	0.528 *	X4∩X7	0.705 **	X12∩X5	0.468 **	X9∩X8	0.584 **
	X9∩X8	0.522 **	X14∩X9	0.703 *	X9∩X8	0.455 **	X10∩X8	0.582 **
	X7∩X5	0.521 **	X4∩X8	0.702 **	X7∩X5	0.453 **	X6∩X7	0.577 *
	X10∩X8	0.521 **	X9∩X4	0.701 **	X9∩X13	0.451 *	X6∩X5	0.569 *
	X3∩X7	0.518 **	X9∩X3	0.692 **	X9∩X6	0.445 *	X2∩X8	0.567 **
2020a	X2∩X7	0.592 *	X4∩X7	0.777 **	X2∩X7	0.531*	X8∩X5	0.726 *
	X1∩X7	0.519 *	X11∩X4	0.734 **	X9∩X8	0.529 *	X10∩X5	0.657 *
	X7∩X5	0.492 *	X7∩X5	0.732 *	X9∩X5	0.519 *	X13∩X7	0.640 *
	X3∩X7	0.476 *	X10∩X13	0.712 *	X9∩X2	0.519 *	X6∩X5	0.634 *
	X10∩X7	0.473 **	X11∩X9	0.712 **	X9∩X7	0.484 **	X8∩X7	0.633 **
	X8∩X7	0.462 *	X10∩X7	0.711 **	X9∩X6	0.477 *	X10∩X6	0.633 *
	X13∩X7	0.460 *	X11∩X7	0.711* *	X2∩X5	0.462 *	X10∩X8	0.628 *
	X9∩X2	0.449 *	X13∩X7	0.710 **	X1∩X7	0.454 *	X6∩X7	0.619 *
	X11∩X7	0.448 **	X9∩X7	0.707 **	X9∩X13	0.453 *	X10∩X13	0.609 *
	X4∩X7	0.446 **	X8∩X7	0.706 **	X4∩X7	0.443 *	X9∩X8	0.592 *

Note: ** and * indicate double factor enhancement and nonlinear enhancement, respectively, and the symbol “∩” represents the interaction between two factors.

## Data Availability

Not applicable.

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
