# Peer review of "Spatiotemporal Variations of Carbon Emissions and Their Driving Factors in the Yellow River Basin"

_ijerph, 2022, doi:10.3390/ijerph191912884_

Round 1
Reviewer 1 Report (New Reviewer)
In the paper "Spatiotemporal variations of carbon emissions and their driving factors in the Yellow River Basin", the authors use a predictive model which incorporates the potential for spatial autocorrelation. Given this significant concern (spatial autocorrelation) is methodologically addressed, and given the detailed analysis and discussion of policy implications, I actually believe that this paper is already suitable for publication in this journal.
Author Response
Please see the attachment

Reviewer 2 Report (New Reviewer)
Dear authors,
thank you for this interesting article. I would suggest to expand the discussion regarding policy implications of your research.
Author Response
Please see the attachment

Reviewer 3 Report (New Reviewer)
Carbon emission reduction in the Yellow River basin plays an important role in global carbon emission and its governance. This study took the Yellow River Basin as an example to analyze the spatial and temporal variation of carbon emissions and its influencing factors. Therefore, this study has important policy guidance values for the high-quality development of the Yellow River Basin. However, as far as the current paper is concerned, there are still some contents that need to be carefully revised. They are as follows:
(1) The abstract of this article lacks the explanation of scientific issues, and the description of research results suggests further summary.
(2) Is "spatial variation; driving factors" in the keyword part too common to reflect the research theme of this article?
(3) What is the scientific problem of this article? The literature review in the second paragraph of the introduction is too simple and lacks a systematic and logical review of the current literature. At present, the author lists the existing literature more and lacks analysis. Therefore, the author's interpretation of the scientific issues in this article is not clear. It is not enough to support the scientific problems of this paper just because of the differences in methods, influencing factors and research areas.
(4) It is suggested to add the evolution map of the spatial pattern of carbon emissions in the Yellow River Basin in the third part (results), that is, the spatial pattern of carbon emissions at different time points, rather than just the cold-hot-spot map.
(5) In sections 4.1 and 4.2 of the discussion, the author's analysis is still weak. The author needs to talk with the existing research through discussion, and find or construct relevant theories to explain the results.
(6) In the discussion part, there is a lack of discussion on policy formulation and implementation.
(7) At present, there are many problems in the language description of this article, which need to be carefully revised.
Round 2
Reviewer 3 Report (New Reviewer)
This article has been greatly improved after modification. Therefore, it is recommended that the author further polish the document and then publish it.
Author Response
Please see the attachment.

This manuscript is a resubmission of an earlier submission. The following is a list of the peer review reports and author responses from that submission.
Round 1
Reviewer 1 Report
The authors have solved all the problems I raised in the previous version.
Reviewer 2 Report
I have carefully read the re-submitted paper for review titled. "Spatiotemporal variations of carbon emissions and their driving factors in the Yellow River Basin". The authors have made several improvements that have convinced me that it is worthy of publication. I therefore recommend its publication in the International Journal of Environmental Research and Public Health. I ask the authors to review the article again and correct some editorial errors, including spaces (missing or redundant), formatting, table splitting, etc.
Reviewer 3 Report
plagiarism software ananlysis showed 16.6 % similarity to other studies.
The manuscript is very long, contains many data and information, but I am missing the aim and the novelty of this study. What is scientific contribution of this study?
What do we have with the outtdated data and the results of the study from 2000-2020?